

# GStatSim V1.0: a Python package for geostatistical interpolation and simulation

Emma J. MacKie [1], Michael Field [1], Lijing Wang [2], Zhen Yin [2], Nathan Schoedl [3,4,5], Matthew Hibbs [3], and Allan Zhang [5]

[1]Department of Geological Sciences, University of Florida, Gainesville, 32601, USA
[2]Department of Geological Sciences, Stanford University, Stanford, 94301, USA
[3]Department of Computer and Information Science and Engineering, University of Florida, Gainesville, 32601, USA
[4]Department of Mathematics, University of Florida, Gainesville, 32601, USA
[5]Department of Statistics, University of Florida, Gainesville, 32601, USA

**Correspondence:** Emma J. MacKie (emackie@ufl.edu)

**Abstract.** The interpolation of geospatial phenomena is a common problem in Earth sciences applications that can be addressed with geostatistics, where spatial correlations are used to constrain interpolations. In certain applications, it can be particularly useful to perform geostatistical simulation, which is used to generate multiple non-unique realizations that reproduce the variability of measurements and are constrained by observations. Despite the broad utility of this approach, there are few open-access geostatistical simulation software. To address this accessibility issue, we present GStatSim, a Python package for performing geostatistical interpolation and simulation. GStatSim is distinct from previous geostatistics tools in that it emphasizes accessibility for non-experts, geostatistical simulation, and applicability to remote sensing data sets. It includes tools for performing non-stationary simulations and interpolations with secondary constraints. This package is accompanied by a Jupyter Book with user tutorials and background information on different interpolation methods. These resources are intended to significantly lower the technological barrier to using geostatistics and encourage the use of geostatistics in a wider range of applications. We demonstrate the different functionalities of this tool for the interpolation of subglacial topography measurements in Greenland.

## 1 Introduction

The interpolation of geological and geophysical observations is a common problem in the geosciences with applications in mineral exploration (Journel and Huijbregts, 1976; Emery and Maleki, 2019), oil reservoir modeling (Kelkar and Perez, 2002; Pyrcz and Deutsch, 2014), groundwater hydrology (Kitanidis, 1997; Feyen and Caers, 2006), and soil sciences (Goovaerts, 1999; Lark, 2012; Xiong et al., 2015). The field of geostatistics emerged in the 1950s with the development of kriging, a method for optimizing spatial interpolation, which was originally used to estimate gold ore grades (Krige, 1951; Cressie, 1990). This theory was expanded by Matheron (1963), who formalized the variogram as a measure of spatial variance. Since then, a number of methods have evolved to include a variety of interpolators and descriptors of spatial statistics (e.g. Cressie and Hawkins, 1980; Solow, 1986; Goovaerts, 1998; Strebelle, 2002).



In contrast to deterministic methods such as kriging, which are designed to minimize estimation variance, geostatistical simulation is designed to reproduce the variability of measurements (Deutsch et al., 1992). Geostatistical simulation involves the generation of many stochastic realizations of a Gaussian random field that retains the spatial statistics of observations. These realizations are conditioned to observed data, meaning they exactly match observations. The ensemble of realizations quantifies uncertainty. This approach is often used to account for subsurface heterogeneity when modeling petroleum reservoirs (Pyrcz and Deutsch, 2014), performing geophysical inversions (Nunes et al., 2012; Shamsipour et al., 2010; Volkova and Merkulov, 2019), and modeling groundwater hydrology (Feyen and Caers, 2006).

While the original geostatistics applications were primarily focused on natural resource exploration, geostatistics applications have expanded to include a wide range of academic topics including climate modeling (Costa and Soares, 2009), natural hazard prediction (Youngman and Stephenson, 2016), and glaciology (MacKie et al., 2020). For instance, geostatistical simulation is used to construct digital elevation models of the topography beneath ice-sheets based on airborne radar measurements (Zuo et al., 2020). These new applications have created geostatistics problems that are larger in scale and use different data types and survey geometries. This means that increasingly flexible and robust geostatistics software are needed to meet these evolving scientific needs.

Despite this expansion of geostatistical simulation to a wide range of geological applications, many existing geostatistical modeling software are primarily intended for use in oil and mineral exploration applications. For example, T-PROGS (Carle, 1999) is intended for use with borehole data, and the Leapfrog geological modeling software is used almost exclusively in industry. The Geostatistics Software Library (`GSLIB` (Deutsch et al., 1992)) test cases are predominantly devoted to modeling porosity and permeability for oil reservoir modeling. While these tools were primarily tested on point measurements such as borehole data, many scientific applications now rely on large-scale airborne geophysical data sets, which typically consist of cross-cutting line surveys that are densely sampled in the along-track direction with large gaps in between survey lines. This data configuration can prove challenging for geostatistical interpolation. Furthermore, geophysical surveys, and remote sensing methods in particular, can generate extremely large volumes of data. This can be problematic for certain software, such as the Stanford Geostatistical Modeling Software (SGeMS, Remy (2005)) which becomes slow to operate and sometimes crashes when users upload large files. As such, many geostatistics software are not directly suited to the scale and nature of certain geophysical data interpolation problems.

The commercial nature of many geostatistics applications and software further restricts the use of geostatistics in academic and educational settings. Many geostatistical modeling software are locked behind a paywall. For example, Leapfrog and T-PROGS are proprietary and cost thousands of dollars per year (e.g. GMS, 2021). Leapfrog and SGeMS only work on Windows operating systems. `GSLIB` is written in FORTRAN, which has a limited user base. These compatibility issues make it difficult to integrate geostatistical simulation with existing scientific workflows. Furthermore, many of these software are intended for use by trained experts and require the purchase of a companion textbook to learn the documentation (e.g. Deutsch et al., 1992; Remy et al., 2009). These accessibility issues create a barrier to producing open-access, reproducible workflows and fail to meet modern accessibility standards such as the FAIR principles (Findable, Accessible, Interoperable, Reusable, (Wilkinson et al., 2016)). Moreover, the restricted availability of geostatistics software limits its use in educational settings. As such,



developments in open-access, user-friendly tools are critically needed for advancing the use of geostatistical simulation in open science and education.

Recently, a growing number of open-source geostatistical modeling tools have become available. This includes the `gstat`

package in R (Pebesma, 2004), and the Python packages `PyKrige` (Murphy, 2014), `GeostatsPy` (Pyrcz et al., 2021), `SciKit-GStat` (Mälicke, 2022), and `GSTools` (Müller et al., 2022). `PyKrige` provides tools for performing kriging interpolation. `SciKit-GStat` offers variogram modeling functions. `GeostatsPy` is a Python wrapper of some of the `GSLIB` functions. `GSTools` provides a number of useful functions including tools for time-series analysis, data transformations, and interpolation. While these resources are important steps towards providing the scientific community with accessible geostatis-

tics software and have significantly expanded the repertoire of geostatistics functions in Python, additional tools are needed for performing geostatistical simulation. In particular, existing Python tools are limited in their ability to accommodate non-stationary, or variations in spatial statistics, and incorporate secondary constraints.

To address the aforementioned needs, we present `GStatSim`, a Python package for performing geostatistical interpolations and simulations. `GStatSim` enables the user to perform a variety of deterministic and stochastic interpolations in Python

including versions of kriging and sequential Gaussian simulation (SGS). Each of these methods is based on the variogram spatial statistics. We include a version of SGS where different variograms can be assigned to different regions. This allows the spatial statistics to vary across a study area. For multivariate problems where secondary data is used to constrain the interpolation, we have provided co-kriging and co-simulation functions. Each of these interpolation algorithms can account for large-scale spatial trends, rotation, and anisotropy.

`GStatSim` is intended to complement existing Python and geostatistics frameworks. While `GStatSim` does not include variogram modeling tools, it is designed to be compatible with existing variogram modeling packages such as `SciKit-GStat`, a robust variogram modeling toolkit with extensive documentation. In contrast to previous tools, `GStatSim` emphasizes interpolation, particularly conditional simulation. We provide new tools for accommodating non-stationarity, or variability in spatial statistics. This package is intended to strike a balance between flexibility and ease of use. The tools are sufficiently

deconstructed to allow for a variety of interpolation workflows, while also limiting the number of parameter selections that must be made. In contrast to previous software that are tested on point data sets, `GStatSim` was constructed with geophysical line data sets specifically in mind.

In addition to providing tools for scientific analyses, `GStatSim` is also intended to be an educational resource. We developed a Jupyter Book that contains a series of tutorials for reproducing the figures in this paper. These tutorials provide intuition on

the different algorithms and guidance on parameter selection. This is intended to make `GStatSim` accessible to non-experts and reduce the amount of time needed to develop models.

In this paper, we provide an overview of `GStatSim` and demonstrate the usage of different features of our package. We also provide a brief description of the underlying theory behind each function. The interpolation functions are demonstrated for the interpolation of ice-sheet bed elevation data measured by airborne ice-penetrating radar. We describe the implementation

considerations and illustrate the advantages and limitations of each interpolation method. We provide an overview of the



**Table 1.** `GStatSim` interpolation functions.

| Function | Description |
|---|---|
| `skrige` | Simple kriging estimator |
| `okrige` | Ordinary kriging estimator |
| `skrige_sgs` | Sequential Gaussian simulation using simple kriging |
| `okrige_sgs` | Sequential Gaussian simulation using ordinary kriging |
| `cluster_sgs` | Sequential Gaussian simulation using simple kriging, where different variograms are defined in different regions |
| `cokrige_mm1` | Co-kriging under Markov Model 1 (MM1) assumptions, uses secondary data as a constraint |
| `cosim_mm1` | Co-simulation under MM1 assumptions, uses secondary data as a constraint |

**Table 2.** `GStatSim` accessory functions.

| Function | Description |
|---|---|
| `grid_data` | Fits conditioning data to a grid of some specified resolution |
| `prediction_grid` | Makes an array of coordinates to be interpolated based on bounding coordinates and resolution |
| `rbf_trend` | Estimates the large-scale trend of the conditioning data using a smoothed radial basis function |
| `adaptive_partitioning` | Breaks conditioning data into blocks based on the density of measurements |
| `find_colocated` | Finds the co-located primary and secondary data points for performing co-kriging and co-simulation |

available user support and documentation. Finally, we discuss possible future directions for further enabling the widespread utilization of geostatistical simulation in academic disciplines.

## 2 **GStatSim** package description

GStatSim is written in Python 3.8 and relies on commonly used packages such as `scipy` (Virtanen et al., 2020), `numpy`
(Harris et al., 2020), `pandas` (McKinney et al., 2010), and `scikit-learn` (Pedregosa et al., 2011). This ensures that GStatSim is easy to use in different computing environments. The source code is written in accordance with PEP8 standards (Van Rossum et al., 2001), making it interpretable and easy to modify.

## 3 Data

We apply the GStatSim functions to the interpolation of ice-penetrating radar measurements of subglacial topography in
Greenland. Subglacial topography is a critical parameter in ice-sheet models (Parizek et al., 2013; Seroussi et al., 2017). For instance, modeled ice flow behavior and subglacial water routing models can be highly sensitive to the method used to



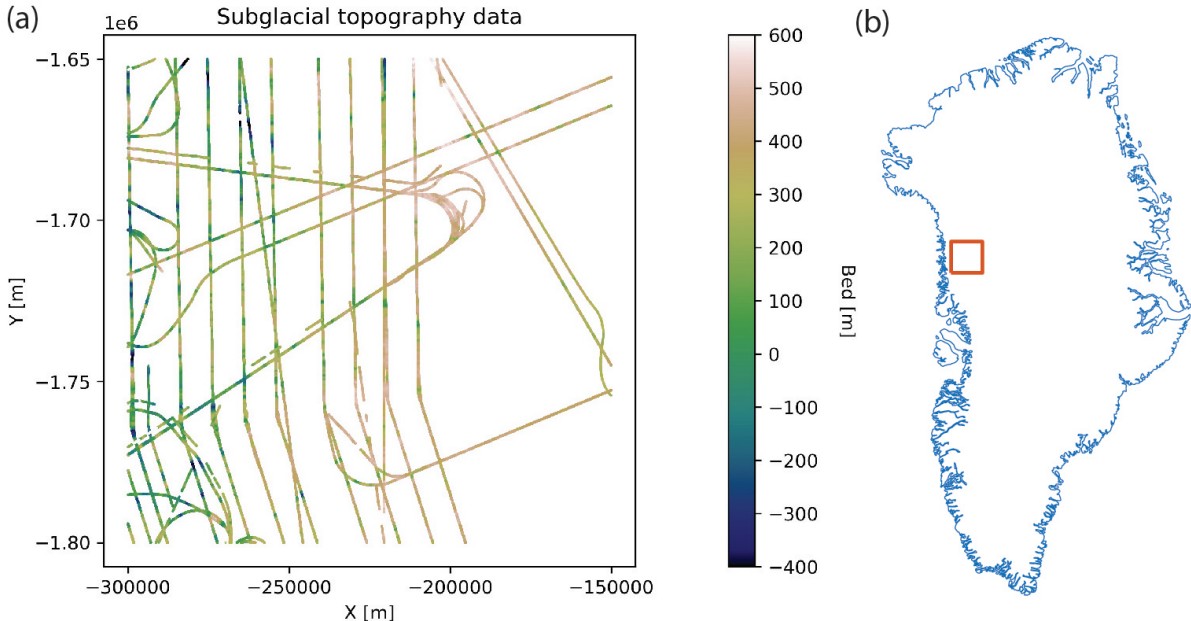

**Figure 1.** Ice-penetrating radar topography data for test region. The coordinates are shown in a polar stereographic projection.

interpolate measurements (MacKie et al., 2021; Wernecke et al., 2022; Law et al., 2022). We use bed elevation measurements from the Center for the Remote Sensing of Ice Sheets (CReSIS, 2022) over a 150x150 km$^2$ region in northwest Greenland (Figure 1). This location was chosen for the variability in line spacing (10-50 km), abundant crossover points, irregular survey

orientations, and the presence of a large-scale trend. These conditions pose a challenge to interpolation, making this data set a good case study for testing the rigor of the `GStatSim` algorithms. For the multivariate applications we use ice surface elevation data from ArcticDEM (Porter et al., 2018) as a secondary source of information for constraining interpolations. We use the 500 m resolution version. The surface and bed elevation coordinates are transformed from geographic coordinates to polar stereographic coordinates using the Antarctic Mapping Tools (Greene et al., 2017) so that Euclidean distances can be

determined.

## 4 GStat-Sim functions and implementation

### 4.1 Data initialization

Prior to performing any geostatistical analysis, we fit the bed elevation data to a 1000 m resolution grid using the `GStatSim` `grid_data` function. We do this because the interpolation examples shown in this paper use a 1000 m resolution. This

way, the spatial statistics of the data will be compatible with the interpolation grid. In order to satisfy the Gaussian assumptions of our interpolation functions, we convert our data to a standard Gaussian distribution using the `scikit-learn`





(a)

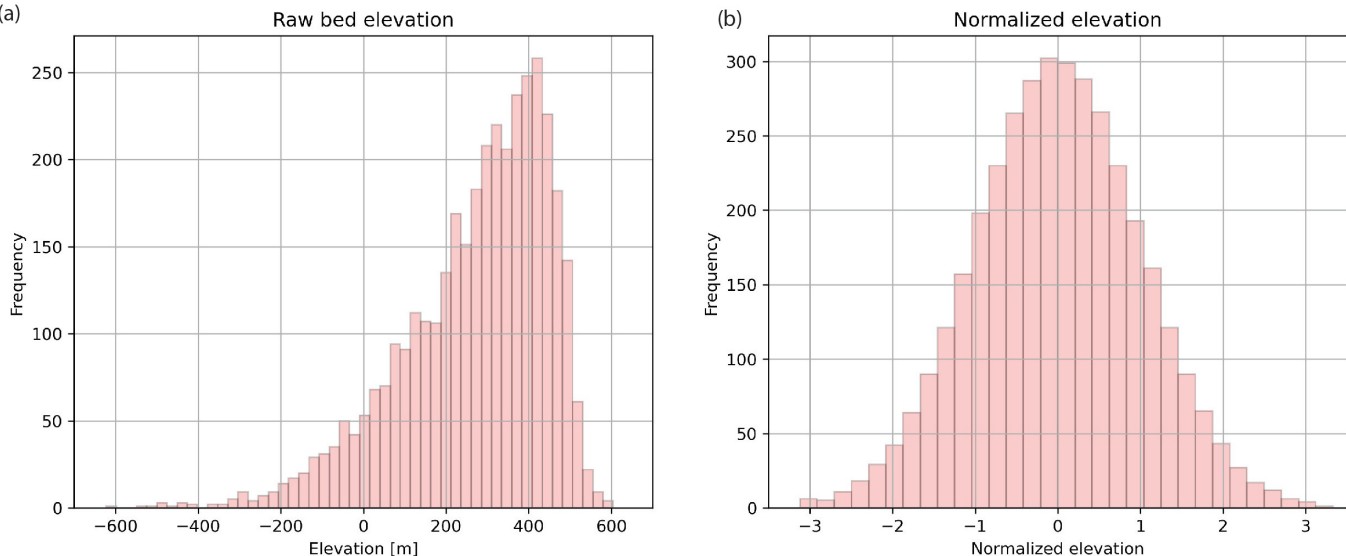

**Figure 2.** (a) Original data distribution. (b) transformed data with a standard Gaussian distribution.

`QuantileTransformer` function. This step converts the data to a standard Gaussian distribution (mean = 0, variance = 1), shown in Figure 2.

## 4.2 Variogram calculation and modeling with SciKit-GStat

The interpolation functions in `GStatSim` rely on the variogram, which describes spatial statistics by quantifying the covariance between data points as a function of their separation distance, or lag (Matheron, 1963). The experimental variogram $\gamma(h)$ is defined as the averaged squared differences of data points separated by a lag distance $h$:

$$\gamma(h) = \frac{1}{2N(h)} \sum_{\alpha=1}^{N} (Z(u_\alpha) - Z(u_\alpha + h))^2 \tag{1}$$

Where $Z$ is a variable, $u$ is a spatial location, and $N(h)$ is the number of pairs of data points for a lag $h$. Typically, the variance increases with lag distance until a threshold is reached where measurements are no longer spatially correlated. This threshold is called the *sill*, which is, in theory, equal to one for normalized data sets. The lag distance where the variogram reaches the sill is called the *range*.

`GStatSim` does not include variogram estimation tools as there are already existing resources (e.g. `SciKit-GStat`). Our interpolation functions are designed to be integrated with outside variogram modeling tools. We perform our variogram analysis with the `SciKit-GStat` functions, which offer robust tools for variogram estimation and modeling. We compute both the isotropic and anisotropic variograms. The anisotropic variogram computes the variogram for different angles from





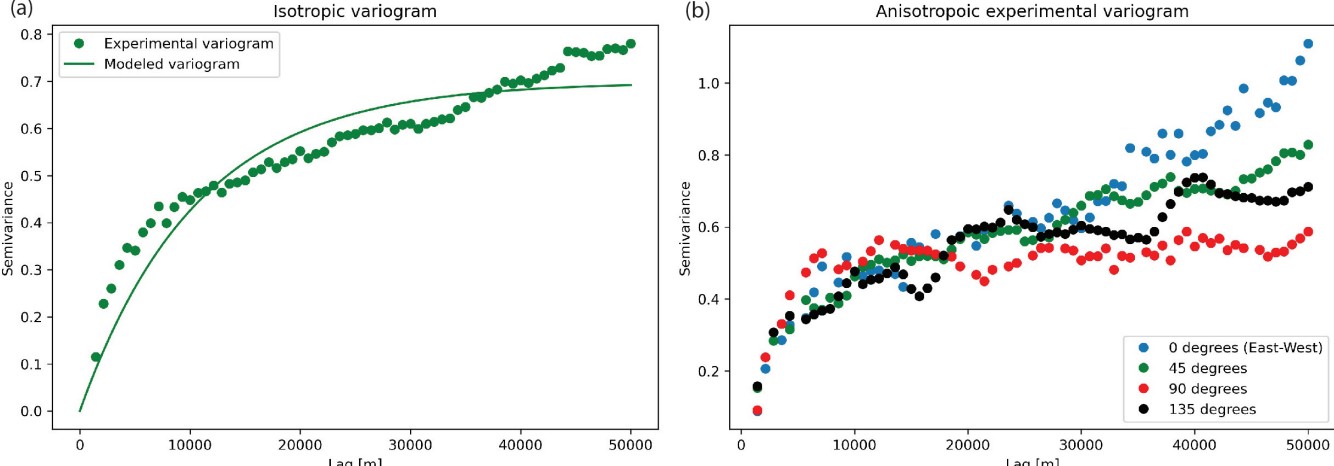

**Figure 3.** (a) Isotropic experimental variogram and variogram model. (b) Anisotropic variogram.

horizontal. Figure 3 shows that the data does not contain significant anisotropy, so only the isotropic variogram is used for the variogram modeling analysis.

In order to use the variogram for interpolation, a variogram model must be fit to the experimental variogram. Although there

are many variogram model types available (e.g. spherical and Gaussian), GStatSim can only accommodate the exponential variogram model:

$$
\gamma(h) = \begin{cases} b + (C_0 - b)\left[1 - \mathrm{e}^{-\frac{3h}{a}}\right] & \text{if } h > 0 \\ 0 & \text{if } h = 0, \end{cases}
\tag{2}
$$

where $a$ is the range parameter, $C_0$ is the sill, and $b$ is the nugget. The nugget describes the variance at near zero lags, which is often the result of measurement error. In most cases, the nugget is zero. The exponential variogram was chosen because it

is a versatile and widely used model (Dubrule, 2017). In practice, variogram model parameters are often selected manually, though SciKit-GStat also offers automatic variogram modeling tools. Using these tools, we fit an exponential model with a nugget of 0, a range of 31.9 km, and a sill of 0.7 for the isotropic variogram. Geostatistical interpolations can be sensitive to the methods and parameters used to compute and model variograms. As such, we recommend referring to the SciKit-GStat documentation for best practices and considerations when conducting variogram analysis.

**4.3  Kriging**

Kriging is used to produce deterministic interpolations of a spatial variable, where the goal is to minimize estimation variance (e.g. Matheron, 1963; Cressie, 1990). For a spatial variable $Z$, kriging models the residuals $Z^*$ from a mean $m$:





$$Z^*(u_\alpha) = Z(u_\alpha) - m(u_\alpha). \tag{3}$$

In cases where the mean is constant, $m$ is treated as a global mean. Each interpolated value $Z^*$ at a location $u$ is the weighted
sum of neighboring measurements:

$$Z^*(u_0) = \sum_\alpha^N \lambda_\alpha Z(u_\alpha), \tag{4}$$

where $\lambda_\alpha$ are the weights on the $N$ data points. These weights account for the variability of the measurements, their proximity
to each other and the node being estimated, and the redundancy between nearby measurements. Specifically, the weights are
determined by solving the kriging system of equations which, for a system with three data points, would look like:

$$\begin{bmatrix} C(u_1 - u_1) & C(u_1 - u_2) & C(u_1 - u_3) \\ C(u_2 - u_1) & C(u_2 - u_2) & C(u_2 - u_3) \\ C(u_3 - u_1) & C(u_3 - u_2) & C(u_3 - u_3) \end{bmatrix} \begin{bmatrix} \lambda_1 \\ \lambda_2 \\ \lambda_3 \end{bmatrix} = \begin{bmatrix} C(u_0 - u_1) \\ C(u_0 - u_2) \\ C(u_0 - u_3) \end{bmatrix} \tag{5}$$

which can be abbreviated to

$$C\lambda = c \tag{6}$$

where $C$ describes the covariance between pairs of data points, and $c$ describes the covariance between each data point and
the location that is being estimated, $u_0$. Note that the covariance is linked to the variogram:

$$C(h) = C(0) - \gamma(h) \tag{7}$$

where $C(0)$ is the variance of the data. We use the modeled variogram parameters determined in Section 4.2.

`GStatSim` contains functions for both simple and ordinary kriging. With simple kriging, it is assumed that the mean is
a constant, known parameter defined by the mean of the data. For data that has undergone a normal score transformation,
the mean is 0. For ordinary kriging, the mean is estimated implicitly within a search neighborhood around the coordinate
being estimated, and simple kriging is applied to the residuals. This makes ordinary kriging more robust to spatial trends. The
ordinary kriging weight estimation procedure is modified from Equation 5 to

$$\begin{bmatrix} C(u_1 - u_1) & C(u_1 - u_2) & C(u_1 - u_3) & 1 \\ C(u_2 - u_1) & C(u_2 - u_2) & C(u_2 - u_3) & 1 \\ C(u_3 - u_1) & C(u_3 - u_2) & C(u_3 - u_3) & 1 \\ 1 & 1 & 1 & 0 \end{bmatrix} \begin{bmatrix} \lambda_1 \\ \lambda_2 \\ \lambda_3 \\ \mu \end{bmatrix} = \begin{bmatrix} C(u_0 - u_1) \\ C(u_0 - u_2) \\ C(u_0 - u_3) \\ 1 \end{bmatrix} \tag{8}$$





where $\mu$ is a Lagrange multiplier. The covariance function is also used to compute the uncertainty, or variance, at each location. The variance of an estimate at location $u_0$, $\sigma_E^2(u_0)$ is defined as:

$$\sigma_E^2(u_0) = C(0) - \sum_{\alpha}^{N} \lambda_\alpha C(u_0 - u_\alpha). \tag{9}$$

The ability of kriging to quantify uncertainty is a major advantage of this method over other deterministic interpolation methods such as spline interpolation.

### 4.3.1 Nearest neighbor octant search

In practice, the covariance matrices in Equations 6 and 8, $C$ and $c$, used to determine kriging weights typically do not use every value in the conditioning data. Instead, it is conventional to use the $N$ nearest neighbors, where $N$ is the number of conditioning nodes (e.g. Pyrcz et al., 2021). This significantly reduces the size of the covariance matrix and ensures that estimates are made using only relevant information. However, the nearest neighbor approach can be problematic when the conditioning data is unevenly spatially distributed. In the ice-penetrating radar example, measurements are sampled densely along flight lines but are entirely absent in between lines. This means that the $N$ nearest neighbors of a coordinate next to a flight line could all be on one side, yielding lopsided conditioning data. This lopsidedness would produce skewed estimates. To avoid this bias, our interpolation algorithms use a nearest neighbor octant search, where the $N$ neighbors are divided amongst the eight different octants of the coordinate plane. This ensures that the conditioning data includes values on all sides of the node being estimated. The ability of the nearest neighbor octant search to produce a balanced conditioning data set is important for producing high quality estimates, particularly for remote sensing data sets where the measurement density is often non-uniform. We note that the primary drawback of this approach is that it increases the algorithm run-time.

### 4.3.2 Covariance matrix inversion

According to Equation 6, solving for the kriging weights $\lambda$ requires the inversion of the covariance matrix $C$:

$$\lambda = C^{-1}c. \tag{10}$$

However, $C$ is not always a positive definite matrix, meaning that $C^{-1}$ cannot be computed. This is especially common for complex scenarios such as ice-penetrating radar data sets which often contain overlapping and conflicting measurements. The matrix singularity issue can be resolved by computing the pseudo-inverse, $C^+$ instead. However the pseudo-inverse requires a time-consuming singular value decomposition with a run-time that scales by $O(n^3)$ with data input size, making it an impractical choice for applications with large data sets. For improved performance, we use the `scipy.linalg.lstsq` function, which computes the least-squares solution to the equation $Ax = b$. Solving for the weights with `lstsq` instead of computing the matrix inversion directly lends greater numerical stability to the interpolation functions. Therefore, all functions in `GStatSim` involving covariance matrix inversions are implemented using `lstsq`.





**Figure 4.** (a) Simple kriging interpolation. (b) Standard deviation for the simple kriging estimate. (c) Ordinary kriging. (d) Difference between simple and ordinary kriging.

### 4.3.3 Kriging implementation and results

We implement simple kriging and ordinary kriging using the `GStatSim skrige` and `okrige` functions, respectively. The input parameters for these functions are the conditioning data, search radius, number of conditioning points, variogram parameters, and a list of coordinates that describe the prediction grid. We use a search radius of 50 km and 100 conditioning

200




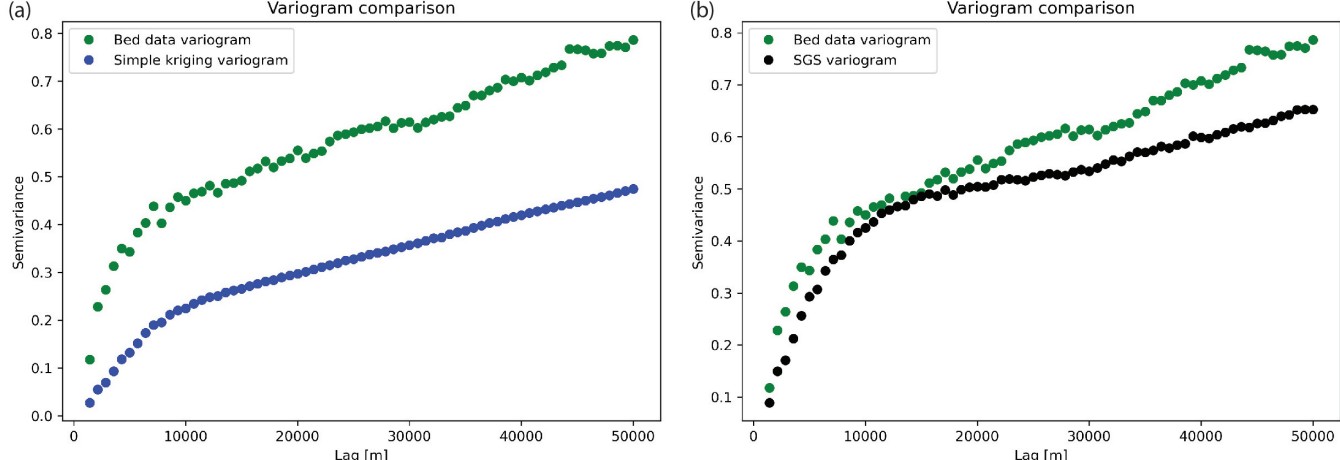

**Figure 5.** (a) Variogram for simple kriging estimate from Figure 4a. (b) Variogram for SGS Realization #1.

points. `GStatSim` includes a `prediction_grid` function that automatically generates a list of coordinates defined by the minimum and maximum coordinate extents and the grid cell resolution. The kriging interpolations are back-transformed using the `scikit-learn QuantileTransformer` function so that the original data distribution is recovered. The kriging results (Figure 4) show that the right side of the ordinary kriging estimate has a slightly higher elevation than the simple kriging estimate. This is because ordinary kriging does not assume that the mean is constant.

To investigate the spatial statistics of kriging interpolation, we compute the variogram for the simple kriging estimate. The resulting variogram, shown in Figure 5a has substantially lower variances than the variogram derived from the data. This highlights an important limitation of kriging interpolation: the variogram is not reproduced, making the interpolated values smoother than observations.

## 4.4 Sequential Gaussian simulation

In contrast to kriging, sequential Gaussian simulation (SGS) is designed to preserve the variance of observations. SGS is the stochastic alternative to kriging. In contrast to kriging where the objective is to optimize local accuracy, SGS is used to generate multiple realizations that reproduce the variogram statistics (e.g. Deutsch et al., 1992). The implementation for SGS is:

1. Define a random path to visit each node.

2. At each node $u$, use kriging to estimate the mean $Z^*(u)$ and variance $\sigma_E^2(u)$.

3. Randomly sample from the Gaussian distribution defined by $Z^*(u)$ and $\sigma_E^2(u)$. This becomes the simulated value at that node.

4. Proceed to the next node and repeat steps 2 and 3 until all nodes have been simulated.







**Figure 6.** (a-d) SGS realizations generated with `okrige_SGS`.

GStatSim has two different SGS functions, `okrige_sgs` and `skrige_sgs` which use ordinary kriging and simple
220  kriging for Step 2, respectively. The input parameters are the same as those used for the `skrige` and `okrige` functions. We
simulate four realizations using `okrige_sgs` using the same input parameters as in Section 4.3.3. To evaluate the ability of





SGS to reproduce the spatial statistics of observations, we compute the variogram for realization #1 (Figure 5). The variogram of the simulated topography matches the variogram of observations. The realizations are shown in Figure 6. Each realization matches the conditioning data and represents an equally probable solution.

## 4.5 Interpolation with anisotropy

Geologic phenomena frequently exhibit anisotropy in their spatial statistics. For example, preferential erosion can cause subglacial topography to be smoother in the direction that is parallel to ice flow (MacKie et al., 2021). The `GStatSim` interpolation functions can account for this anisotropy by accepting major and minor range parameters. The major range is the range of the variogram in the smoothest (major) direction. And the minor range is the range of the variogram in the minor direction, which is orthogonal to the major direction. Not all pairs of data points are oriented along the major or minor orientations, so the range is treated as the radius of a rotated ellipse where the major and minor ranges are the lengths of each axis, and the degree of rotation is determined by the angle of anisotropy. The major angle orientation is defined as the angle from zero (horizontal) of the major direction. In practice, these parameters can be determined by modeling the variogram in multiple directions and determining the major angle through visual inspection.

To demonstrate the implementation of interpolation with anisotropy, we perform simple kriging and SGS with anisotropy for angles of 0° and 60°. We use the modeled variogram parameters from previous sections (nugget = 0, range = 31.9 km, sill = 0.7). We use 31.9 km as the minor range and set the major range to be 15 km greater than the minor range for exaggerated effect. These interpolations (Figure 7) have clearly visible anisotropy.

## 4.6 Mean non-stationarity

Data sets often contain variations in spatial statistics, a condition known as non-stationarity. Mean non-stationarity occurs in the presence of a large-scale trend or systematic changes across a study area. While ordinary kriging can accommodate variations in the mean, to some extent, it does not extrapolate well (Journel and Rossi, 1989). For this reason, it is sometimes convenient to detrend the data prior to variogram modeling and interpolation. Detrending has the additional advantage of producing residual data that more closely represents a random Gaussian process, which makes the variogram modeling more robust. In contrast, data with a strong trend can yield an experimental variogram that never reaches an obvious sill, making it difficult to fit a variogram model.

Here we demonstrate kriging and SGS with a trend (Figure 8). This is achieved by:

1. Estimate the trend.

2. Subtract the trend from the data to obtain residual measurements.

3. Apply a normal score transformation to the residual data.

4. Conduct a variogram analysis on the transformed residual measurements.

5. Interpolate the residuals.





**Figure 7.** (a) Simple kriging with anisotropy at 0 degrees. (b) Simple kriging with anisotropy at 60 degrees. (c) SGS with anisotropy at 0 degrees. (d) SGS with anisotropy at 60 degrees.

6. Back-transform the interpolation to recover the original data distribution.

7. Add the back-transformed interpolation to the trend.





**Figure 8.** (a) Trend estimate. (b) Simple kriging of detrended data. (c) The simple kriging estimate from (b) added to the trend from (a). (d) The residual bed elevation data after the trend is subtracted from the data. (e) SGS applied to the detrended data. (f) The SGS interpolation from (e) added to the trend from (a).

While it is common practice to estimate the trend by fitting a linear or polynomial function to the data (e.g. Neven et al., 2021), determining higher-order polynomial models for complex landscapes can be difficult. Instead, we estimate the trend using radial basis functions (RBF), which are functions that change with distance from a location (Broomhead and Lowe, 1988). The RBF trend estimation was implemented using the GStatSim rbf_trend function, which relies on the scipy RBF implementation. The rbf_trend function input parameters are the gridded conditioning data, the grid cell resolution, and a smoothing factor. The smoothing factor ensures that the trend is not overfitting to measurements. The smoothing parameter should be at least as large as the largest data gap. We use a smoothing factor of 100 km.

To evaluate the performance of the detrended SGS approach, we compute the experimental variogram for residual data (Figure 8d) and the residual SGS (Figure 8e). The variograms, shown in Figure 9, show close agreement between the spatial statistics of the residual measurements and simulation. Note that both variograms reach a sill of 1, in contrast to the variogram



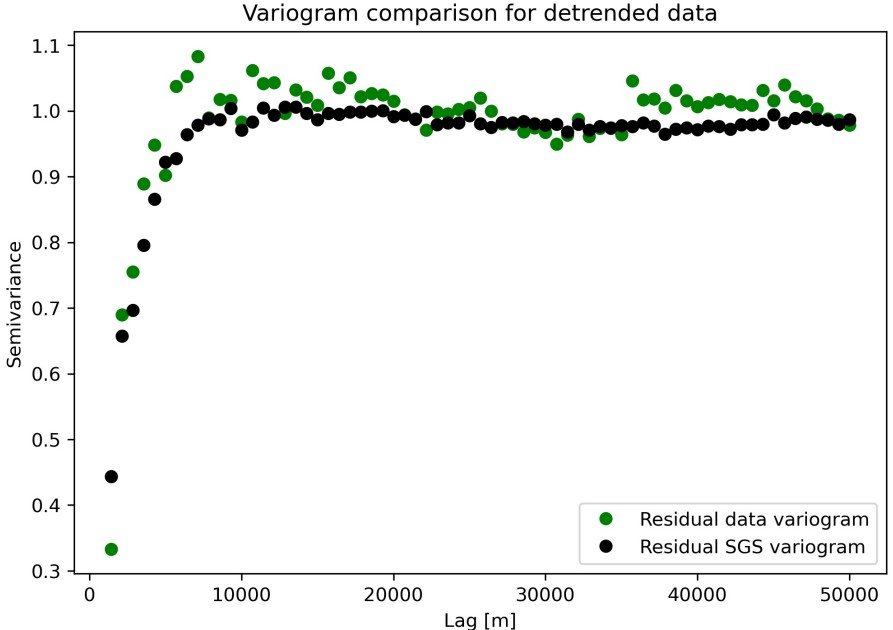

**Figure 9.** Variogram for detrended data from Figure 8d and simulation of trend from Figure 8e.

265  for the data without detrending (Figure 5). According to the theoretical defintion of the variogram (Equation 2), data with a standard Gaussian distribution should reach a sill of 1. As such, the detrended data is better suited to variogram-based interpolation.

## 4.7 Variogram non-stationarity

Variogram non-stationarity describes the condition when the variogram statistics vary throughout a domain. For example,
270  the topographic roughness varies throughout Greenland (Cooper et al., 2019). To handle these cases, GStatSim includes a `cluster_sgs` function where the user can assign different variograms to different areas. It is implemented by partitioning the data into different clusters, and then computing the variogram for each cluster. To simulate a grid cell:

1. Choose a random grid cell.

2. Find the nearest neighbor value. Look up the cluster number of the nearest neighbor point.

275  3. Using the variogram parameters associated with the cluster of the nearest point, simulate a value with SGS.

4. Assign the nearest neighbor cluster number to the simulated value.

5. Repeat until each grid cell is simulated.



This process ensures that the boundaries between different clusters vary for each realization. `cluster_sgs` takes the same inputs as SGS as well as the cluster numbers for each data point, and a dataframe with the variogram parameters for each cluster. Users can use any method they like to partition the data into different clusters. In the next two subsections, we outline two approaches.

### 4.7.1 K-means clustering approach

One of the simplest ways to cluster data is with k-means clustering, where the data is partitioned into $k$ clusters such that the average squared sum of distances from the center within each cluster is minimized. We apply k-means clustering to the conditioning data and determine three clusters based on the coordinates and bed elevation (Figure 10a). The justification for using these parameters is that data points that are geographically close to each other and have similar elevation values should have similar spatial statistics.

We apply the variogram analysis to the data in each cluster separately using the procedure described in Section 4.2. To make the differences between clusters more pronounced, we manually modify several of the variogram parameters to exaggerate the differences in spatial statistics for visualization purposes. Clusters 0 and 2 are assigned angles for anisotropy (45° for Cluster 0 and 90° for Cluster 2). The minor ranges for Clusters 0 and 2 are assigned based on the automatic variogram modeling. The major ranges are set by adding 15 km to the minor ranges. Cluster 1 is given a sill of 0.6, and clusters 0 and 2 are assigned sills equal to 1. The `cluster_sgs` simulation using these parameters produces visible differences in spatial statistics throughout the domain. For example, the anisotropy is clearly visible in the region defined by Cluster 0 (Figure 10b). Despite the extreme differences in parameters used for each cluster, the cluster boundaries do not have artifacts or inconsistencies.

### 4.7.2 Adaptive partitioning

The k-means clustering approach described above provides a simple approach to partitioning the data, but the decision boundaries can seem arbitrary and result in clusters with widely varying sample counts. To overcome these challenges we present a recursive implementation of a divisive, density based clustering approach we call Adaptive Partitioning. The algorithm begins by treating the dataset as a single cluster and quartering this cluster into four rectangles of equal area. For each subsequent cluster, the areal extent of the cluster continues to be quartered until the number of data samples contained within the cluster is below the maximum number of samples allowed or the minimum cluster size has been reached. `GStatSim` implements this as a fully recursive function named `adaptive_partitioning` with primary parameters *max_points* which controls the maximum points allowed in a cluster and *min_length*, which controls the minimum side length of a cluster. It may take tuning of these two parameters to produce a desirable partitioning, including adjusting *max_points* to create clusters of adequate sizes given the measurement density. *min_length* should be greater than the expected variogram range to ensure that there is sufficient data to accurately model the variogram parameters. Figure 11a shows the clusters resulting from `adaptive_partitioning` with *max_points* of 800 and *min_length* of 25 km.

The variogram parameters for each cluster are modeled automatically and then modified to show exaggerated differences between clusters. Cluster 1 is modified to have a sill of 0.6. Clusters 6 and 12 are given 90° and 45° directions of anisotropy,





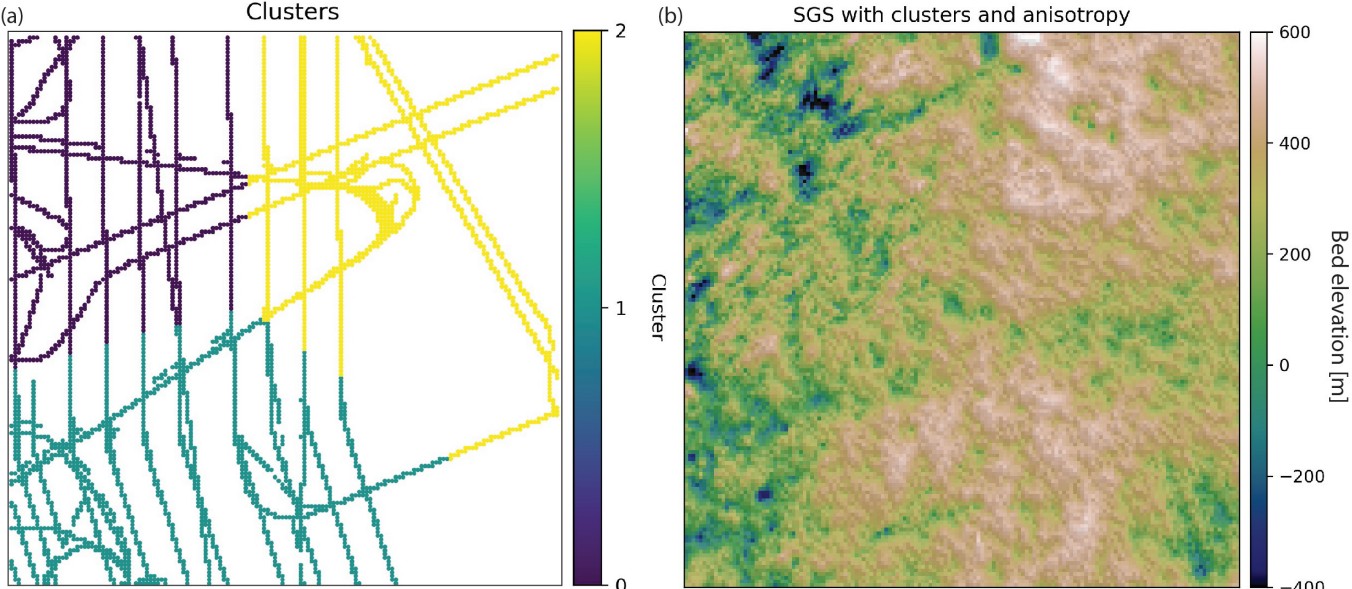

**Figure 10.** (a) K-means clustering of conditioning data. (b) SGS using different variogram parameters for each cluster.

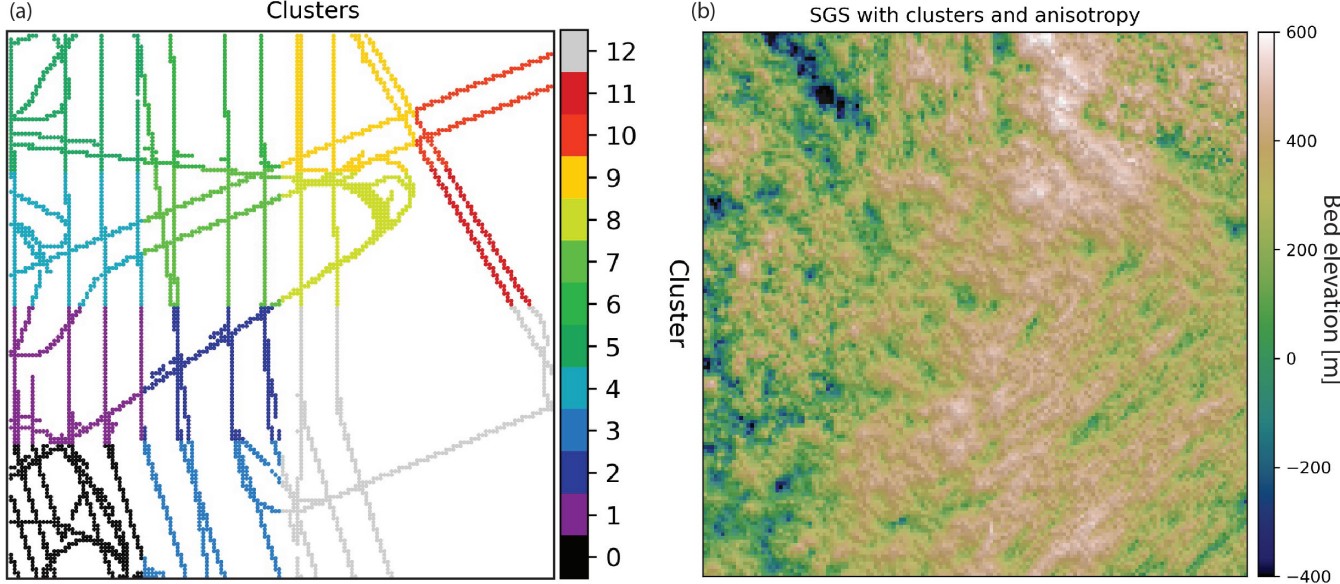

**Figure 11.** (a) Ice surface elevation data. (b) Normalized ice surface elevation measurements that are co-located with radar measurements.
(c) Co-kriging interpolation. (d) Co-simulation.




respectively, with major ranges that are determined by adding 15 km to the modeled range. These regional differences in variogram parameters are apparent in the resulting simulation in Figure 11.

## 4.8 Co-kriging and co-simulation

Secondary information can be integrated into interpolations via co-kriging and co-simulation, where a well-sampled secondary
variable is used to constrain the interpolation of a sparsely-sampled primary variable. For example, if the ice surface elevation (Figure 12a) is found to be correlated with the subglacial bed elevation, then the ice surface can be used to improve the interpolation. We note that the ice surface and subglacial topography are unlikely to be linearly related (Ng et al., 2018); this example is merely intended to demonstrate the usage of our co-kriging and co-simulation tools. In theory, co-kriging and co-simulation are implemented by modeling the variogram for each variable and a cross-variogram that describes the covariation
between the variables. In practice, the full co-kriging system of equations is difficult to solve so approximations are used instead (Almeida and Journel, 1994; Journel, 1999).

While there are several different ways to implement co-kriging (Journel, 1999), GStatSim only includes functions for co-located co-kriging under Markov Model 1 (MM1) assumptions, as described by Almeida and Journel (1994). This version was chosen for its simplicity; expert geostatistics knowledge is not needed to use this algorithm. Co-located co-kriging means that
an estimate is made using only the secondary data point that is *co-located* with, or at the same location as, the grid cell being estimated. This significantly reduces the number of conditioning data points that are used to estimate each grid cell. Co-kriging with MM1 approximates the cross-variogram through a Markov assumption of conditional independence. It is assumed that a secondary data variable $Z_2$ at a location $u$ is conditionally independent of the primary variable $Z_1$ at a location of $u'$, given $Z_1(u)$. $u'$ refers to a spatial location that is not $u$. This statement of conditional independence is written as:

$$E(Z_2(u)|Z_1(u) = z_1, Z_1(u') = z_1(u')) = E(Z_2(u)|Z_1(u) = z_1), \tag{11}$$

where $E$ refers to the expected value. In the geostatistics community, this Markov assumption is referred to as Markov Model 1 (Almeida and Journel, 1994). See (Journel, 1999; Shmaryan and Journel, 1999) for other cokriging models. In the case where the ice surface elevation is used as a secondary constraint, MM1 means that the expected value of the ice surface at a location $u$, given knowledge of the bed elevation at location $u$, is independent of the bed elevation value at other locations. Under MM1
assumptions, the cross-correlogram describing the covariation between the primary and secondary variables is calculated as

$$\rho_{12}(h) = \rho_{12}(0)\rho_1(h) \tag{12}$$

where $h$ is the lag distance and $\rho$ is the correlogram, calculated by:

$$\rho(h) = 1 - \gamma(h) \tag{13}$$





assuming that $\gamma(h)$ is the normal scores variogram. While the variogram measures spatial covariance, the correlogram

measures spatial correlation. This means that $\rho_{12}(0)$, the correlogram at a lag distance of zero, is simply the correlation coefficient between the primary and secondary variables. Because the cross-covariance between the primary and secondary data relies only on the correlation coefficient and primary variogram, the secondary variogram is not needed. This means that co-kriging and co-simulation with MM1 only require the primary variogram and the correlation coefficient to determine the co-kriging weights. The simple co-located co-kriging estimate $Z_1^*$ at a location $u_0$ is defined as

$$Z_1^*(u_0) = \sum_{\alpha}^{N} \lambda_\alpha Z_1(u_\alpha) + \lambda_2 Z_2(u_0), \tag{14}$$

where $\lambda_\alpha$ are the weights on the $N$ primary data points, $Z_2(u_0)$ is the co-located secondary datum, and $\lambda_2$ is the weight for the secondary datum. This means that the co-kriging estimate of a bed elevation value is the weighted sum of bed elevation data points within a specified search radius and the co-located ice surface elevation datum. The variance is computed using equation 9. Co-SGS is implemented in the same way as SGS, except that Equation 14 is used instead of Equation 4.

To implement co-kriging and co-SGS, we use the variogram model parameters from Section 4.2 and compute the correlation coefficient, $\rho_{12}(0)$, between the bed elevation measurements and co-located ice surface elevation. For convenience, we developed a `find_colocated` function that extracts the co-located data points from the secondary variable (Figure 12b). $\rho_{12}(0)$ was determined to be 0.6. The interpolations are performed using the GStatSim `cokrige_mm1` and `cosim_mm1` functions. These functions require the same input parameters as `skrige`, `okrige`, `skrige_sfs`, and `okrige_sfs` with

the addition of $\rho_{12}(0)$ and the secondary data set.

The co-kriging and co-SGS results are shown in Figure 12. Compared to the previous kriging and simulation examples, these interpolations produce visibly higher elevation topography on the right side of the region. This is due to the correlation with the ice surface. To evaluate the simulation performance, we compute the correlation coefficient between the ice surface elevation (Figure 12a) and co-simulated bed elevation (Figure 12d). This correlation coefficient is found to be 0.7, which is greater

than the correlation coefficient of the co-located data, 0.6. This inflated correlation between the primary and secondary data is a known issue with MM1 (Shmaryan and Journel, 1999). Co-simulation with MM1 is also known to produce interpolation artifacts (MacKie et al., 2021) and artificially high variance (Journel, 1999). This is because the MM1 assumption screens out the influence of the primary data on the secondary data, except at the location $u_0$, which can lead to the underestimation of the redundancy between the primary and secondary variables. For this reason, MM1 is most effective in situations where the

primary variable is smoother than the secondary variable, which is not the case in our example. As such, it is important to check the statistics (variogram and correlation) of simulations produced with `cosim_mm1` to ensure that this method is used appropriately.





**Figure 12.** (a) Ice surface elevation data. (b) Normalized ice surface elevation measurements that are co-located with radar measurements. (c) Co-kriging interpolation. (d) Co-simulation.



## 5   Tutorials and documentation

Each of the previous interpolation examples can be reproduced in tutorials found in the `GStatSim` GitHub page
(https://github.com/GatorGlaciology/GStatSim), Jupyter Book (https://gatorglaciology.github.io/gstatsimbook/intro.html), or
Zenodo repository (https://doi.org/10.5281/zenodo.7274640). These tutorials can be downloaded as Jupyter Notebook files
and are intended to enhance educational pathways to geostatistics.

## 6   Installation, licensing, and redistribution

GStatSim is added to the Python Package Index, commonly known as PyPi, and can be installed via the *pip install gstatsim*
command. It can also be installed directly from GitHub with *git clone https://github.com/GatorGlaciology/GStatSim*. This
package has an MIT license making it free to use and redistribute without restriction.

### 6.1   Discussion

Many existing geostatistical software are not open-source or directly suited to large-scale remote sensing problems. The
`GStatSim` package provides an accessible, user-friendly alternative to these software. It relies on existing packages (e.g.
`SciKit-GStat`) where possible and can easily be integrated with existing Python tools and scientific analyses. The tuto-
rials and documentation provide ample scaffolding for users without prior geostatistics experience. This will encourage the
utilization of geostatistical simulation in new scientific domains and enhance educational pathways in geostatistics.

The bed topography interpolation examples show many different ways to interpolate the same data set, each with differ-
ent assumptions, advantages, and limitations. Many of these approaches can be combined. For example, `cluster_sgs`,
`ckrige_mm1`, and `cosim_mm1` could be applied to detrended data. In practice, the optimal interpolation method or combi-
nation of methods will depend on the structure of the data and the scientific objectives.

Although the focus of this manuscript is subglacial topography, the `GStatSim` routines could be applied to numerous geo-
scientific problems. Cryosphere applications could include the interpolation of englacial layers (Born and Robinson, 2021), ma-
rine sediment cores, and radiometric properties of subglacial conditions (e.g. Chu et al., 2021). Beyond glaciology, `GStatSim`
could be used for downscaling, potential field data analysis (Volkova and Merkulov, 2019), and planetary applications such as
improving the resolution of digital terrain models (Gwinner et al., 2009). The `cluster_sgs` function would be particularly
useful for capturing regional changes in spatial statistics, and the nearest neighbor octant search makes `GStatSim` well-suited
to geostatistical modeling with geophysical line data. Furthermore, the ability of co-kriging and co-simulation to include sec-
ondary constraints provides a mechanism for synthesizing multiple geophysical and geological data sets. `GStatSim` could
also be integrated with geophysical modeling routines such as SimPEG (Cockett et al., 2015) to perform inversions. Specifi-
cally, geostatistical simulation can be used in Markov chain Monte Carlo frameworks for solving inverse problems (Haas and
Dubrule, 1994; Oh and Kwon, 2001; Nunes et al., 2012).

Each of the example interpolations in this paper takes 2-3 minutes to run on a MacBook Pro with an Apple M1 chip. The run times increase exponentially with size and resolution improvement. Most of this computational time is attributed to the nearest
neighbor octant search. As such, geostatistical simulation can become cumbersome and time-intensive for large ensembles, depending on the simulation grid size, nearest neighbor search radius, and number of conditioning points. There are many approaches for accelerating SGS (Dimitrakopoulos and Luo, 2004; Daly et al., 2010; Mariethoz, 2010; Nussbaumer et al., 2018), though these methods lack flexibility or require greater computational resources. Future work is needed to develop tools with improved performance while maintaining the flexibility and functionality. GStatSim could also be modified to include
additional variogram models and interpolation methods such as indicator kriging (Solow, 1986).

## 7 Conclusions

Spatial interpolation is a common problem in the geosciences, though the availability of open-access software remains a key barrier to the widespread utilization of geostatistics. The GStatSim package addresses this barrier by making available Python tools and educational materials for implementing various interpolation methods. The tools here could be applied to a variety
of geological and geophysical data sets for both research and educational purposes and can be integrated with existing Python tools and workflows. The inclusion of non-stationary SGS, co-kriging, co-simulation, and trend estimation functions make GStatSim particularly useful for applying geostatistics to complex phenomena.

*Code and data availability.* The source code, data, and tutorials are permanently archived in Zenodo (https://doi.org/10.5281/zenodo.7274640). These materials can also be accessed from GitHub (https://github.com/GatorGlaciology/GStatSim), PyPi (https://pypi.org/project/gstatsim/),
and our Jupyter Book (https://gatorglaciology.github.io/gstatsimbook/intro.html).

*Author contributions.* EJM is the primary developer of GStatSim and author of the manuscript. MF made major contributions to the package and writing. LW, ZY, NS, MH, and AZ contributed to the GStatSim software development.

*Competing interests.* The authors have no competing interests to declare.

*Acknowledgements.* We would like to thank Eric Stubbs for providing technical support on the software development. We thank Fernando
Peréz for his guidance on the direction of this repository. We thank Caroline Riggall, Leo Ramsey Watson, and Caleb Koresh for helping test the tools. This work was sponsored by the Earth Sciences Information Partners (ESIP) Lab and the National Science Foundation (NSF) GeoSMART program.



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
