# Peer review of "GStatSim V1.0: a Python package for geostatistical interpolation and conditional simulation"

_EGUsphere, 2022_

## Author Response (AR1)

The authors present a well-written, concise and easy to understand paper about a new Python package called GStatSim. The package is designed to fill the gap of missing geostatistical simulations tools in the scientific Python landscape, as identified by the authors.

The presented package meets modern programming and publishing standards as common in Python and I personally like the approach of focusing educational use-cases. The provided tutorials are outstanding.

With some minor comments, concerning the paper and the code itself, I am looking forward to see this nice work published.

Best,

Mirko Mälicke

We would like to thank Dr. Mälicke for their thoughtful and insightful comments. We have revised our manuscript and code to address these concerns. The main changes we made were the addition of spherical and Gaussian variogram models, and additional discussion on how GStatSim compares to GSTools. We have also added a statement in the introduction clarifying that in this paper, we are not recommending a specific interpolation method for this case study; we are merely demonstrating how different interpolation algorithms are applied. In the discussion, we provide additional guidance on usage considerations.

In fact I really only have one main point:

- P.7 135. My personal main comment to the work is the limitation to exponential variogram models. I think the authors should make clearer, why other models are not supported. Comparing to other geostatistical software, this design decision seems to make integrations with other packages difficult, as most of them share a substantial range of available models.

  In case there are algortihmic, technical or scientific limitations that I am not aware of right now, it would be most helpful to discuss these limitations in the discussion transparently and beyond the simple outlook, that more models could be implemented, as currently stated in the discussion.

We agree that including other variogram models would expand the usage of our tools. As such, we have added spherical and Gaussian variogram modeling options. We have added Figure 9 (Section 4.5) demonstrating the application of these models. The variogram model comparison is demonstrated for a synthetic example because the Gaussian variogram is a very poor fit for topography (Figure 3).

Minor comments:

- P.3 L.62-67. I think this paragraph could make clearer, why the approach of introducing a new package from scratch was chosen, over the extension of existing packages, as ie. GSTools already has one field simulation function.

I have added a brief section (Section 3) where we tried to apply GSTools, PyKrige, and GeostatsPy to our topography data set. Unfortunately, these programs crash due to memory issues when I use PyKrige and GSTools. (We have done a lot of troubleshooting with GSTools and can only get it to work for very small problems). Our case study has ~4000 conditioning points, whereas the GSTools conditional interpolation demos only have 5 points. I'm guessing that GSTools just wasn't designed with large conditioning data sets in mind. GSTools works well for unconditional simulations.

One of the other differences is that GStatSim uses a nearest neighbor octant search, which improves the interpolation and simulation quality when interpolating lines of data. We have added a figure (Figure 4). The other difference with GStatSim is that we have simulation functions that can account for regional variations in spatial statistics and incorporate secondary constraints (cokriging and co-simulation).

- P.3 L.68-74. GStatSim implements ordinary and simple kriging, which also exists is various other Python packages. What makes the two new implementations different from existing ones?

We couldn't get the PyKrige and GSTools kriging functions to work for our case study. GStatSim is robust when using large, messy data sets.

Algorithmically, the main kriging difference is that GStatSim uses a nearest neighbor octant search, which improves the quality of interpolations of line surveys. Aside from that, our simple kriging and ordinary kriging functions don't add much novelty. However, since our other functions build on kriging, we felt it was worth keeping them in the package for consistency.

We have added more discussion on the differences between GStatSim and previous tools.

- P.3 L.80-82. GStatSim was designed with geophysical point data in mind. Does this affect the application on 'classic' point based data? Are there differences in the algorithms, ie. for kriging?

GStatSim works with point-based or non-gridded data as well. We have added an example of this (Figure 9).

- P. 4. L.94 To me this reads like GStatSim is written for Py3.8 only, but I am confident it will run on newer versions as well. This could be checked systematically (see my last comment) and adjusted accordingly.

Yes, it works for Python 3.0-3.10. We have been using GStatSim with newer versions. We have modified this statement (line 106).

- P.5-6 L.113-118: For me it is not completely clear, if the described procedures are part of GStatSim and its functions, or if these transformations are part of the presented workflow and specific to the presented dataset.

This is a good point. All of our interpolation algorithms assume that the data has a Gaussian distribution, but it is not strictly necessary for our algorithms to run. We have added clarification (line 117).

- P.8 L.162ff, the ordinary kriging algorithm is described to be robust to spatial trends. In case spatial trends are expected, why not using universal kriging in these cases?

Universal kriging is just simple kriging with a trend, which is a perfectly viable option. This is done in section 4.6 (and we have added a sentence at line 266 saying that kriging with a trend is also known as universal kriging). However, trend estimation requires some assumptions (especially if using polynomial trends). So ordinary kriging is sometimes preferred with an unknown constant mean. In the geostatistics community, ordinary kriging vs. universal kriging seems to be a matter of personal preference, so we make both options available.

I won't go too deeply into the theory in the manuscript, but if you're curious, this paper does a nice job of explaining the differences:

Journel, A. G., & Rossi, M. E. (1989). When do we need a trend model in kriging?. *Mathematical Geology*, *21*, 715-739.

- P. 9 section 4.3.1: As I was completely unfamiliar with NN octant search, further references to the method, and especially a description of the '8 main octants of the coordinate plane' would be highly appreciated. In addition: Does this 'alignment' have advantages for use-cases, other than the geopyhsical data described?

We have added a schematic comparing the NN octant search with a simple NN search (Figure 4) and added a reference (line 187).

- P.11 L.201-205: Similar to my comment to P.5-6, I also think that here it would be again helpful to state more clearly if the described back-transformation is part of the presented kriging algorithms, or a workflow step specific to the presented data.

We have modified this line (now 208) to say:

"After GStatSim interpolation is applied, the kriging estimates are back-transformed using the scikit-learn QuantileTransformer function so that the original data distribution is recovered."

- P.11 section 4.4: In my opinion, this whole section could be substantially upgraded by the figure from the documentation demos: 4. Sequential Gaussian simulation, which illustrates the algorithm:

https://raw.githubusercontent.com/GatorGlaciology/GStatSim/f03710188fb200985c86d65832dd7f8e68c1d662/images/SGS_cartoon-01.jpg

We have added this figure. This figure is adapted from MacKie et al., (2021) which was published as Gold Open Access (https://www.igsoc.org/publications/journal-of-glaciology/copyright-notice). In order to meet the terms of reuse and redistribution, we include the following statement in figure caption:

"This figure is modified from MacKie et al., (2020), which was published under a Creative Commons Attribution license (https://creativecommons.org/licenses/by/4.0/)."

- P.11 L.214 Is 'path' referring to an actual path here? Is the sequence at which nodes are simulated limited by the constrain that the nodes have to be neighboring, which is my understanding of a path? If yes, why?

The simulation path refers to a random simulation order and is a commonly used term in geostatistical simulation (e.g. Nussbaumer et al., 2018). The nodes don't need to be neighboring or connected. For clarification we have modified line 226 to say "Define a random simulation order to visit each node."

Nussbaumer, Raphaël, et al. "Accelerating sequential gaussian simulation with a constant path." *Computers & Geosciences* 112 (2018): 121-132.

- P.12-13 L. 219-224: I think it would be beneficial to give the user a short insight on why there are two different sgs algorithms and when you would choose which one in general terms.

We have added a statement to line 233:

"The results are unlikely to differ significantly between these two methods, but okrige_sgs may be more robust when large-scale trends are present."

- P.15 L. 255-261: Are the authors advising to apply the rbf_trend function to any dataset, before applying GStatSim functions or are there also downsides of the approach the user needs to be aware of? In addition t would be helpful to provide references to more detailed descriptions of the RBF approach, as some readers (including myself) might not be familiar with it.

We recommend using rbf_trend over polynomial trends, but we don't believe that trends are always necessary. In some cases it may be helpful to detrend with RBF, but that will depend on the scientific application and nature of the data. We have added references to the RBF method.

- P.16 -17 section 4.7: Are there general rules, how each grid cell can be attributed to a cluster? Is this based on expert knowledge? How should I decide on the number of clusters necessary?

This should be based on expert or geological knowledge. For example, some sort of geological boundary could be used to delineate clusters. We have added the following line to section 4.8.1:

"We use a k of 3, which was chosen arbitrarily for demonstration purposes. In real applications, external information or domain expertise may be used to inform the clustering decision."

- P. 17 section 4.7.1: I guess this relates to my previous comment: How do I know that I need three clusters here? The presented split of the dataset in Figure 10a) seems quite arbitrary to me.

Yes, this was an arbitrary decision. We could have used a different number of clusters. Here we are just demonstrating the method. In reality, there may be outside information or domain knowledge that could be used to inform this decision. The clusters could also be determined by other methods, or grouped manually. We could have also used bed elevation as an input in the clustering (lower elevation areas may be smoother because of sediment infill). Ultimately, this decision process will be unique for different applications. The cluster_sgs function doesn't care how the area is clustered.

- P. 17 section 4.72.: This sections adresses the issues I raised in my last comment, but still, the decision of cluster size is for me as abritrary as the decision on the number of clusters. Ultimately, by setting (spatial) cluster sizes I will also imply a narrow range of how many clusters can be found, right? Maybe I miss something. I think it could be beneficial to add a short description of why the specified parameters were chosen in the style of an example. I am sure that I will get it then.

The adaptive partitioning approach was created to determine a less arbitrary alternative to k-means clustering, and has been previously used in Yin et al., (2022) (we've added a citation to this reference). The length size should be big enough to be able to compute the range. The isotropic variogram model computed in section 4.2 had a range of 32 km, each cluster should be big enough to be able to capture this. As such, we chose a length of 25 km, which has a diagonal of ~35 km (25*sqrt(2)). Technically, any method could be used to group data into zones with different variograms. The cluster_sgs function just requires cluster labels on each of the data points, but the algorithm doesn't care how those clusters were determined.

Yin, Z., Zuo, C., MacKie, E. J., & Caers, J. (2022). Mapping high-resolution basal topography of West Antarctica from radar data using non-stationary multiple-point geostatistics (MPS-BedMappingV1). *Geoscientific Model Development*, *15*(4), 1477-1497.

- P. 19 L.322-329: How does the described procedure relate to kriging with external drift? Can that be used as well, to incorporate secondary variables into an interpolation or simulation?

Kriging with external drift can incorporate secondary data by using it as a trend, which is sometimes a good option. They're fairly different approaches. Sometimes one method is more accurate than the other (e.g. Boezio et al., 2006), or is better at reproducing the spatial statistics when doing simulation (depending on the application).

Boezio, M. N. M., Costa, J. F. C. L., & Koppe, J. C. (2006). Kriging with an external drift versus collocated cokriging for water table mapping. *Applied Earth Science*, *115*(3), 103-112.

Finally, there are a few questions/comments concerning the code itself. Non of them are part of the review in a strict sense:

- Why are some methods implemented as class instance methods, while no of the classes has an implemented constructor function (__init__)?

  If no attributes/settings/data are shared across the methods, why use class instance methods? I am asking more out of curiosity, as I can't see any advantages beside a grouping of related functions (which can be achieved with sub-modules as well).

The main reason we grouped the functions is to be in compliance with PEP8 style guidelines, and to make it easier for ourselves when we expand upon this package.

- The docstrings seem to be of non-stand format (I think). More as a suggestion: Why not use ie. Numpy styled docstrings and create a sphinx-based documentation?
  - https://numpydoc.readthedocs.io/en/latest/format.html
  - https://www.sphinx-doc.org/en/master/usage/quickstart.html
  - https://docs.readthedocs.io/en/stable/intro/getting-started-with-sphinx.html

We have updated the docstrings to use this format.

- The package seems to lack unittests. Although these bring some substantial extra-work with them, I personally always made the experience that this is well invested effort (Many, many developers simply ignore code without tests). I know there are different approaches out there, but from my point of view reliable code is as important as FAIR. In python there are frameworks like pytest (https://docs.pytest.org/en/7.2.x/) that make it particularly easy to add a few tests that ie. run the full analysis given the nice data already included. With CI tools like GH actions, Travis CI or circle CI, the tests will have zero overhead once implemented and can run on all current Python versions and operation systems in parallel. Please take this comment more as my personal opinion and suggestion, to dramatically increase the range of this really nice software.

We appreciate this suggestion and recognize the importance of performing unittests. We haven't gotten to it yet, but it's one of our priority items as we continue to expand GStatSim.
* * *
Review of "GStatSim V1.0: a Python package for geostatistical interpolation and simulation" by E. MacKie et al.

Joseph A. MacGregor

10 February 2023

This manuscript introduces a Python package for geostatistical interpolation and simulation and provides several use cases for Greenland radar-sounding data. The authors present a natural progression introduced of geostatistical complexity and each example is carefully justified, along with the overall motivation for this package, which I expect could be widely used. This MS is exceptionally well-written and is among a very small number that I've reviewed whose key messages were fully understood after a single reading. I perused the Jupyter notebooks and found them similarly informative. I have only minor comments below.

We thank Dr. MacGregor for the supportive and constructive feedback. We have revised the manuscript to address these concerns, with particular attention to improving figure clarity and aesthetics.

While the MS (including the Introduction) are very clear and well-motivated, the Introduction is a bit long-winded (10 paragraphs) and could be more concise.

We cut the introduction down to 8 paragraphs.

The MS weaves back and forth between meters and kilometers for length units, mostly using meters in figures and kilometers in text, but not exclusively. I suggest using kilometers only for figures and text…fewer zeroes for the reader to parse.

We have modified the figures and text to use kilometers for length scales, and meters for elevation.

Throughout the MS, I suggest a discrete color map instead of continuous for the gridded bed topo examples. Perhaps a 50 m interval. Little information would be lost, but I suspect it would be easier to discern the patterns referred to in the text and difference between various methods, particularly examples like Figure 6.

We understand the reasoning here, but personally prefer to use continuous colormaps for continuous variables.

Figure 1b: Label full Greenland plot and blue coastline. Would be nice to have an ice-sheet outline also.

We have modified Figure 1b to show the ice-sheet outline.

107: ArcticDEM is now out of date compared to GrIMP (https://nsidc.org/data/nsidc-0645/versions/1), but that is a minor issue and does not warrant recalculation.

Given the modest correlation between the bed and ice surface, the updated ice surface DEM is unlikely to affect the results. Nevertheless, we appreciate the reviewer bringing this to our attention and will incorporate GIMP into any future case studies requiring ice surface constraints.

109: It's a bit odd, given the FAIR stance of GStatSim, to use a MATLAB package (AMT) for projection of geographic coordinates, which could be done with GDAL instead.

FAIR point! We have modified this to use the NSIDC polar stereographic conversion tools (line 96).

113: From this sentence, I don't fully understand what operation is done. Looking at the Jupyter Notebook, it seems like the data are binned (averaged within those bins?) to 1 km. A bit more exposition needed here.

Yes, exactly. We average values within bins. We have added clarification to line 113.

Figure 3a: Label nugget/sill/range values mentioned on 142.

We chose not to label variogram parameters because Figure 3 now includes multiple variogram models, but we have modified line 142 (now line 145) so that it refers to Figure 3a.

132: How is it assessed that there is no significant anisotropy? Seems like they disagree at lag > ~35 km, but that is above sill.

The anisotropy at large (>35 km) lags is due to trend effects rather than roughness. The range distance (maximum correlation length) is ~30 km, so variogram differences at lags greater than 30 km won't have an effect on the interpolation. We have added the following statement to Section 4.2 for clarity:

"While the anisotropic variograms do exhibit some disagreement at lags greater than ~35 km, these differences can be attributed to regional trends in the topography. As such, only the isotropic variogram is used for the variogram modeling analysis. Additionally, the first part of the variogram with small lags is more important than the tail distributions when it comes to calculating the kriging weights"

Figure 4b,d: Use different color maps for what are fundamentally different quantities being shown compared to 4a/c. I suggest a red/blue colormap for 4d centered on zero.

That makes sense. We have updated the color maps for Figures 5 (previously 4), 9, and 15.

388: Permit me to not-so-humbly suggest that MacGregor et al. (2015; doi: 10.1002/2014JF003215) is a more relevant reference here, as that study performed ordinary kriging to interpolate englacial layers that could be improved using GStatSim and is closer to the examples discussed in the MS.

We agree that this is a more appropriate reference. This has been updated.